# Sorafenib in Metastatic Papillary Thyroid Carcinoma with *BRAF K601E* Mutation on Liquid Biopsy: A Case Report and Literature Review

**DOI:** 10.3390/medicina58050666

**Published:** 2022-05-17

**Authors:** Marianna Caterino, Mario Pirozzi, Sergio Facchini, Alessia Zotta, Antonello Sica, Giorgio Lo Giudice, Raffaele Rauso, Elisa Varriale, Fortunato Ciardiello, Morena Fasano

**Affiliations:** 1Division of Oncology, Department of Precision Medicine, University of Campania, 80010 Naples, Italy; caterinomarianna@gmail.com (M.C.); mario.pirozzi@unicampania.it (M.P.); sergio.facchini.93@gmail.com (S.F.); alessiazotta@libero.it (A.Z.); antonello.sica@fastwebnet.it (A.S.); fortunato.ciardiello@unicampania.it (F.C.); 2Maxillofacial Surgery Unit, Department of Neurosciences, Reproductive and Odontostomatological Sciences, University of Naples “Federico II”, Via Pansini, 5, 80131 Naples, Italy; giorgio.logiudice@unina.it; 3Head and Neck Unit, Clinica Cobellis, 84078 Vallo della Lucania, Italy; raffaele.rauso@unicampania.it; 4Division of Oncology, Fatebenefratelli Hospital, 80123 Naples, Italy; elisava@libero.it

**Keywords:** differential thyroid cancer (DTC), TKI, sorafenib

## Abstract

Differentiated thyroid cancer (DTC) includes papillary and follicular carcinomas and is the most common type of thyroid cancer. The incidence of this cancer has increased in the last few years, and even if its prognosis is generally good for a subset of patients that does not respond to radioactive iodine (RAI) therapy, the prognosis is much worse: the median overall survival (OS) from discovery of metastasis is 3–5 years and the 10-year survival rate is only 10%. Several mutations, including RAS or RET, as well as BRAF signaling, are associated with thyroid cancer. Liquid biopsy may be useful in selected patient to identify genomic alterations and thus allowing for a precision medicine approach with target therapy. Sorafenib, an oral multi-kinase inhibitor, can be used in the treatment of DTC. Case presentation: A 77 years old. man with diagnosis of metastatic DTC and evidence of presence of mutation of BRAF K601E on liquid biopsy was treated with sorafenib, showing a good response to the treatment and an improvement in the quality of life (QoL). Currently, this patient is still on treatment with sorafenib, gaining control of a multi-metastatic disease, generally characterized by a very poor prognosis. In conclusion, sorafenib has an active role in the treatment of DTC. It also has been considered the standard of care for patients with advanced unresectable hepatocellular carcinoma (HCC) and renal cell carcinoma (RCC). In our case we observe the efficacy of using sorafenib in Papillary thyroid carcinoma (PTC) such as confirming both stable disease (SD) in the CT scan as clinical benefit with an increase in QoL. Therefore, use of sorafenib remains an important treatment option, even in case of BRAF mutation, despite a rapidly evolving treatment landscape. It also seems important to perform liquid biopsies, especially in patients in whom it is not possible to repeat a new tissue biopsy. Ongoing clinical trials continue to evaluate sorafenib in different settings, and in combination with other therapies in DTC and HCC.

## 1. Introduction

Thyroid cancer is the most common endocrine malignancy. Thyroid parenchyma consists of two main cell types, thyroid follicular cells giving rise to differentiated thyroid carcinoma (DTC), and parafollicular or C cells giving rise to medullary thyroid carcinoma (MTC).

DTC includes papillary and follicular carcinomas, and it is the most common type of thyroid cancer, accounting for nearly 90–95% of all malignant thyroid tumors; MTC, instead, accounts for about 1–2% of all thyroid cancers [1].

Papillary thyroid carcinoma (PTC) accounts for approximately 80–90% of all thyroid cancers; it is often more aggressive in elderly female patients, being responsible for 96% of all new deaths. Generally, it shows up as a solitary nodule, and less frequently as multi-focal lesions [2]. It can contain branched papillae that possess a fibrovascular peduncle covered by one or more layers of cuboid epithelial cells. Foci of lymphatic invasion are often present, while invasion of blood vessels is relatively rare, especially in smaller lesions. Metastases to satellite cervical lymph nodes are estimated to occur in about half of all cases.

The incidence of thyroid cancer has rapidly increased in recent years. It appears to be associated with increases in PTC, and is likely a reflection of improved imaging technologies resulting in earlier detection and diagnosis [3].

The prognosis for patients with DTC is generally good, with treatment options including surgical resection, hormone treatment and radioactive iodine (RAI, also called Iodine-131) therapy [3]. However, for a subset of patients who go on to develop metastatic DTC (mDTC) that does not respond to RAI therapy, the prognosis is much worse: median overall survival (mOS) from discovery of metastasis is 3–5 years and the 10-year survival rate is only 10% [3,4].

Growth factors play an important role in carcinogenesis; these include vasculoendothelial growth factor (VEGF), often expressed in thyroid tumors [4,5] and the mutation of murine sarcoma viral oncogene homolog B (BRAF), which is found in about half of PTC with a frequency of 45%, where it is associated with tumor recurrence, more rapid cancer growth and a higher death rate [4,6]. BRAF V600E is less widespread in anaplastic thyroid cancer (ATC), where it has been reported in about 25% of cases and is rare in follicular thyroid cancer (FTC). In the context of PTC, BRAF mutation rarely co-occurs with genetic alterations within rat sarcoma (RAS) or rearranged-during-transfection (RET) genes. This allowed the reclassification of PTC into two molecular subtypes, called BRAF-like and RAS-like, which are each associated with a different cancer risk. In addition, both preclinical and clinical evidence suggests that the presence of BRAF V600E may reduce the absorption of RAI therapy. At the molecular level, another driver mutation found in PTC is the reorganization of RET [6].

The prevalence of RET rearrangements in PTC varies considerably between different studies (2.5–73%) [6,7] but RET rearrangements are associated with a more favorable prognosis [6]. Finally, RAS mutation and RET/PTC translocation, resulting in aberrant signaling of BRAF [5], which appear to play a role in the pathogenesis and/or progression of mDTC, are targeted by sorafenib, a tyrosine kinase inhibitor (TKI), which represents a rational therapeutic option in DTC.

Sorafenib is an oral, multi-kinase inhibitor with established single-agent activity in many tumor types [8,9]. It was initially identified as a Raf inhibitor; it later became clear that sorafenib has a number of targets, including several protein kinases in the Ras-Raf-MEK-ERK signaling cascade (Figure 1) and various oncogenic mutations of Ras and Raf, including the BRAF V600E mutation associated with tumor angiogenesis and invasion, and the silencing of tumor-suppressor genes in multiple cancer types [10,11,12,13]. Despite its inherent selectivity, sorafenib can cause several adverse events (AEs); most frequently, toxicities included skin toxicities (rash, pruritus, hand–foot skin reaction (HFSR)), fatigue, weight loss, gastrointestinal toxicities (diarrhea, dyspepsia, mucositis) and cardiovascular toxicity (hypertension).

Liquid biopsy is a low-cost, non-invasive alternative to surgical biopsies which is frequently used to detect circulating tumor cells, DNA and exosome and to identify mutation in cancer genomes. While firstly used for its prognostic value, liquid biopsy role has been recently shifted to diagnosis and therapy with identification of specific mutations in chosen genes [14].

Here, we report the results of treatment with sorafenib in one of our patients treated at the Oncology Department of the University L. Vanvitelli with a rapidly progressive K601E BRAF PTC, previously treated with unsatisfying results with surgery and RAI. We used liquid biopsy to detect multiple mutations and evaluate possible targeted therapies.

## 2. Case Description

In December 2020, a 77-year-old man, without any family history of neoplastic pathology and only hypertension and prostatic hyperplasia as his medical history, performed an ultrasound scan after onset of edema of the neck, deep stridor, dyspnea at rest and persistent cough, with evidence of a multi-nodular goiter and severe tracheal compression. On 12 December 2020, the patient underwent a total thyroidectomy. Histological examination deposed for well differentiated, multi-centric, bilateral, RET wild-type papillary thyroid carcinoma, with wide solid follicular architecture areas, signs of microscopic parathyroid extracapsular extension, presence of vascular invasion and clear margins. The pathological stage was pT2, and post-surgery CT scan in January 2021 showed no signs of distant metastasis. Thus, in March 2021, the patient underwent radioactive iodine therapy (RAI), with ablative purpose (single dose of 1850 mBq of 131-i (50mCi)).

In April 2021, following the appearance of widespread bone pain and difficulty in walking, which required the support of crutches, the patient performed a whole-body scintigraphy with I-131 and 18-FDG total body PET. Both instrumental investigations evidenced a pathological accumulation of radiocompound to the cervical and abdominal lymph nodes, of both hemithorax, at the basin, and several bone metastases (D11 body erosion, D6, D9 involvement and in the left acetabular roof). In the same month, the patient performed a total-body CT scan that showed the presence of metastases both at bone level (presence of 46x34 mm of inhomogeneous tissue that infiltrates the occipital bone expanding to the subcutaneous tissue, pressing on the transverse venous sinus) and at the lung level (multiple subcentrimetric bilateral nodules). It confirmed, moreover, the bone lesions already known, with evidence of other lesions at the level of C3-C5-C6, left iliac wing and D12. (Figure 2).

Blood exams reported the following results: FT3 = 3.2 pg/mL; FT4 = 14.8 pg/mL; TSH = 8.29 µU/mL; Thyroglobulin = 9852 ng/mL; Calcitonin = 5.7 pg/mL.

In addition, the patient performed an MRI of the vertebral column (Figure 3) in order to assess the compression on the spinal cord, which confirmed the already known bone picture.

The patient only came to our attention for the first time in April 2021; the colleagues who previously had him as their charge had already suggested antalgic radiotherapy on the known secondary bone lesions (finished on April 30th–10 sessions, total dose 30 Gv). The patient complained of pain in the spine, great walking impairment to the point of needing a wheelchair, intense asthenia, dizziness and dyspnea due to mild-intensity efforts.

Considering the instrumental investigations already carried out, the metastatic sites and the symptoms, neurosurgical and orthopedic consultations were prescribed to evaluate possible maneuvers on the cervical column, reporting no surgical indication at the moment, and maxillofacial counseling to allow the start of the therapy with Denosumab. In addition, we prescribed antalgic therapy with opioid drugs. The patient practiced liquid biopsy in our institute with next-generation sequencing, which reported K601E BRAF mutation and TERT promoter mutation. TERT promoter mutations have been more frequently reported in undifferentiated or anaplastic thyroid carcinomas, but also in 11.3% of papillary thyroid carcinomas, and has been linked with increased aggressiveness and mortality, especially in combination with V600E BRAF mutations; in PTC, TERT promoter mutation has been found more frequently in patients with V600E BRAF mutation.

Nevertheless, due to the rapidly worsening condition of the patient, no off-label application for any antiBRAF TKIs was possible, and at the beginning of May 2021, the patient started a first-line treatment with sorafenib at the standard dose: two tablets of 200 milligrams in the morning and two of 200 milligrams at night, every day, in addition to Denosumab.

After three months of treatment, due to the appearance of hematological toxicity (grade 2 thrombocytopenia) and grade 2 skin toxicity (HFSR), we reduced the dose of sorafenib to three tablets per day, in addition to the specific symptomatic therapy prescription.

At the same time, the patient reported significant improvement in previously complained symptoms, with total regression of dyspnea, cough, asthenia and pain. For the latter, in fact, the patient suspended the previously prescribed antalgic therapy. In addition, the patient experienced restitutio ad integrum of the walking impairment, no longer needing a wheelchair or walking crutches. The total-body CT scan of August 2021 (Figure 4(A1,A2)) showed stable bone disease and complete lung and lymph node response. In addition, blood chemistry confirmed the benefit shown on the CT scan with a significant decrease in thyroglobulin values (13,510 vs. 25,460.00 ng/mL). In view of the clinical benefit and the marked response to the CT scan, the patient continued treatment as previously prescribed.

In December 2021, due to the appearance of mucositis with swallowing difficulty, it was necessary to further modify the schedule (one week three tablets a day and the next, two tablets a day) with good treatment tolerability. The last total-body CT scan performed in January 2022 (Figure 4(B1,B2)) confirmed the maintenance of the response already highlighted in August 2021.

## 3. Discussion

Thyroid cancer is a rare type of cancer that develops in the thyroid gland, a part of the endocrine system. Globally, it represents about 1–4% of all malignant tumors with a continuously increasing incidence, both for the intensification of screening programs and for the use of more sensitive diagnostic tools that can identify even small tumors.

About 90% of malignant thyroid tumors are DTC, 80% of which is represented by the FTC. Usually, DTC can be treated effectively with surgery, RAI and L-thyroxine therapy. However, 7–23% of patients develop distant metastasis, showing a refractory to RAI therapy [15]. These patients have poor prognosis, and the absence of effective therapy (including chemotherapy) makes their clinical management difficult [16]. In fact, before the introduction of sorafenib among the options for treatment, there was no standard of care for these patients.

Knowledge of the molecular biology of thyroid cancer has increased over the past decade and much attention has been paid to the continued search for genetic alterations.

In detail, dysregulation of multiple signaling pathways, such as mitogen-activated protein kinase (MAPK), phosphoinositide-3-kinase (PI3K), receptor tyrosine kinase (RTK) and Wingless/Integrated (WNT), has been reported to contribute toward the pathogenesis of thyroid cancer and is generally associated with the genetic alterations of genes involved in these pathways [17]. The most common driver genetic alterations observed in DTC include mutations in BRAF, RAS and RET/PTC rearrangements; in PTC and ATC, tumor protein p53 (TP53) mutations are often observed; and in MTC, point mutations in RET oncogene and pathogenic RAS variants are common. These genetic alterations are affecting MAPK and PI3K signaling [18].

This molecular landscape of thyroid cancer stressed the importance, as a necessity, of using novel target therapy in this disease. The molecular therapies currently approved for refractory and aggressive thyroid cancer are Lenvatinib and Sorafenib, both of which been shown to improve progression-free survival (PFS); their use is also characterized, not infrequently, by AEs including diarrhea, palmar–plantar erythrodysesthesia and hypertension, which require careful monitoring. Therefore, in a balance between benefit and toxicity in the use of approved drugs, it becomes important to be able to identify molecular markers that can guide in the choice of oncological therapies.

In recent years, there has been a progressive and growing interest in the role of liquid biopsies in thyroid cancer (Figure 5).

Indeed, blood is an organic fluid that is readily available, and from which it is possible to obtain various pieces of clinically relevant information. This represents a risk-free, low-cost, non-invasive method and an alternative to surgical biopsy allowing detection, if present, of multiple tumor markers and mutations [19]. Indeed, finding tumor samples often requires invasive procedures and it is indeed not feasible to recollect samples to monitor response and/or identify resistance mutations. Liquid biopsy represents a way to bypass these problems and allows for quick and less aggressive molecular profiling. The main applications of liquid biopsy in thyroid cancer are differential diagnosis, mutational screening and monitoring response/resistance.

It is important to point out that, while the potential diagnostic role of liquid biopsy appears to be recognized in some cancers such as lung cancer, its role remains controversial in thyroid cancer [19]. In our case, although it was only few months after surgery and thus a recent tumor specimen was already available, we chose to perform liquid biopsy for practical reasons: surgery was performed in a different hospital in another city, increasing the paperwork and prolonging the time needed to retrieve a tissue block and analyze it in our institute, and with the rapidly worsening clinical condition of the patient, liquid biopsy offered a non-invasive and faster (turnaround time about 10 days) alternative to obtain the same information. As initially foreseen, we could get ahold of the block only after treatment was already started. 

A retrospective trial by Lin et al. showed that, in lung adenocarcinoma, tissue-based NGS had significantly higher sensitivity than plasma-based NGS (94.9% vs. 52.6%, although not all patients in this setting were treatment naïve) [20], whereas Liebs et al. found a concordance between tumor tissue and cfDNA of 63%, 55% and 11% in colorectal cancer, melanoma and head and neck squamous cell carcinoma [21]. Recently, Khatami, F. et al. reported the importance of liquid biopsy in both diagnosis and prognosis. In fact, more studies have correlated the presence in blood of BRAF mutation with the early detection of thyroid cancer [22]: the BRAFV600E mutation is detected in approximately two-thirds of all differentiated thyroid cancers and is strongly recommended for risk stratification in PTCs. Circulating tumor cells (CTCs) count has been also linked to prognosis, with a higher count related to presence of metastasis or poor response to RAI [23].

Because of this, one might imagine a role for liquid biopsy as an alternative to tissue-based NGS in cases where tumor specimens are not easily available or biopsy cannot be performed. Tissue analysis may be then used as a confirmatory test of a negative results, since an insufficient sample is virtually indistinguishable from a negative sample.

Circulating tumor DNA (ctDNA) can also be used potentially to plan targeted therapy, evaluate treatment response, analyze mutational load at scheduled intervals, and screen for new mutations that potentially can lead to treatment resistance, predating eventual disease progression: for example, it has also been demonstrated that a higher amount of circulating BRAF V600E during treatment with antiBRAF is related to worse overall survival [19].

BRAFV600Eis the predominant mutation in PTC (18–87%), but has also been detected in PDTC, ATC and HTC [24,25]. Other rarely detected BRAF variants include BRAFK601E mutation, large insertion/deletions, fusions and small deletions [26]. The most common RAS mutations occur at codons 12, 13 and 61, which are considered mutational hot-spot regions and may affect any of the three RAS genes—HRAS, KRAS and NRAS. The most predominant RAS mutation to be detected in thyroid neoplasms is NRAS61 [27]. *BRAF* and *RAS* mutations are mutually exclusive genetic alterations, suggesting that the presence of either of these genetic variants is sufficient for thyroid tumorigenesis [28].

B-Raf is a serine/threonine-specific protein kinase. It a member of the Raf kinase family, which plays a role in regulating the MAP signaling pathway of kinases/ERKs, sending signals within cells that are involved in the division, differentiation and secretion of cells. Since 2002, it has been known that the BRAF protein has been mutated into several human tumors and, to date, more than 30 mutations have been identified. The most frequent mutation (90%) is V600E, in which valine (V) is replaced by glutamic acid EUR at codon 600. These mutations can lead to constitutive activation of BRAF with consequent activation of the MEK and ERK pathway downstream and uncontrolled cells proliferation.

In view of this, B-Raf inhibitors are potential therapeutic candidates for BRAF-mutated tumors, and were first tested in the most common mutation, BRAFV600E.

Sorafenib has activity against B-Raf and C-Raf and is also indicated for the treatment of advanced unresectable hepatocellular carcinoma and in renal cell carcinoma.

It was the first TKI to be approved by the Food and Drug Administration (FDA) for the treatment of progressive mDTC refractory to RAI therapy, in November 2013. The approval was based on results of DECISION trial, a randomized, double blind, placebo-controlled, phase 3 study which investigated sorafenib (400mg orally twice daily) in patients with RAI-refractory locally advanced or mDTC that had progressed within the past 14 months. This trial showed that sorafenib significantly improved PFS compared with placebo (10.8 months vs. 5.8 months) with an overall tolerable profile of toxicity. After that, in February 2015, a second TKI, Lenvatinib, another multi-kinase inhibitor, was approved in the same setting based on SELECT trial, a randomized, double-blind, multi-center phase III study, the involving 261 patients with progressive DTC. Lenvatinib was associated with a longer median PFS of 18.3 months vs. 3.6 months in the placebo group.

In the clinical case described by us, the presence of the BRAF mutation identified through the liquid biopsy and the possibility to choose only between two prescribed therapeutic options, Lenvatinib and sorafenib, made the therapeutic decision fall on sorafenib considering its inhibitory potential on BRAF.

Obviously, as the patient came to our observation in severe clinical conditions, the timing that would have required an off-label prescription or nominal use of more specifically anti-BRAF drugs such as vemurafenib and dabrafenib, which are already widely used in other BRAF-positive tumors, would not have been compatible with the strong symptoms complained of by the patient and which, in fact, required a rapid and immediate therapeutic intervention. This, therefore, did not allow us to consider such therapeutic alternatives.

## 4. Conclusions

Our case is one of the few cases reported in literature showing exceptional response to Sorafenib in a BRAF K601E patient with a high metastatic burden. We used a liquid biopsy to detect mutations, as it was not feasible to retrieve surgical tissue and the patient’s poor performance status made it impossible to repeat a biopsy, and we identified the need to include research on the BRAF mutation in the molecular panel of thyroid carcinomas. Literature data on BRAF mutations, especially in PTC since, suggest a link between K601E mutations, and better prognosis and the encapsulated follicular histological variant, as described for our patient. Finally, the presence of the BRAF mutation may be responsible for determining sensitivity to sorafenib [29,30].

In our case, in fact, we have observed an astonishing effectiveness of sorafenib in PTC, as confirmed by both the CT scan and the improvement of clinical conditions and quality of life of the patient. Although clinical studies on thyroid carcinoma have shown conflicting results regarding the correlation between BRAF Class 2 activating mutations and the efficacy of panRAF inhibitors, such as sorafenib, the use of sorafenib remains an important treatment option, despite a rapidly evolving treatment landscape. In fact, ongoing clinical trials continue to evaluate sorafenib in different settings and in combination with other therapies in HCC and DTC.

## Figures and Tables

**Figure 1 medicina-58-00666-f001:**
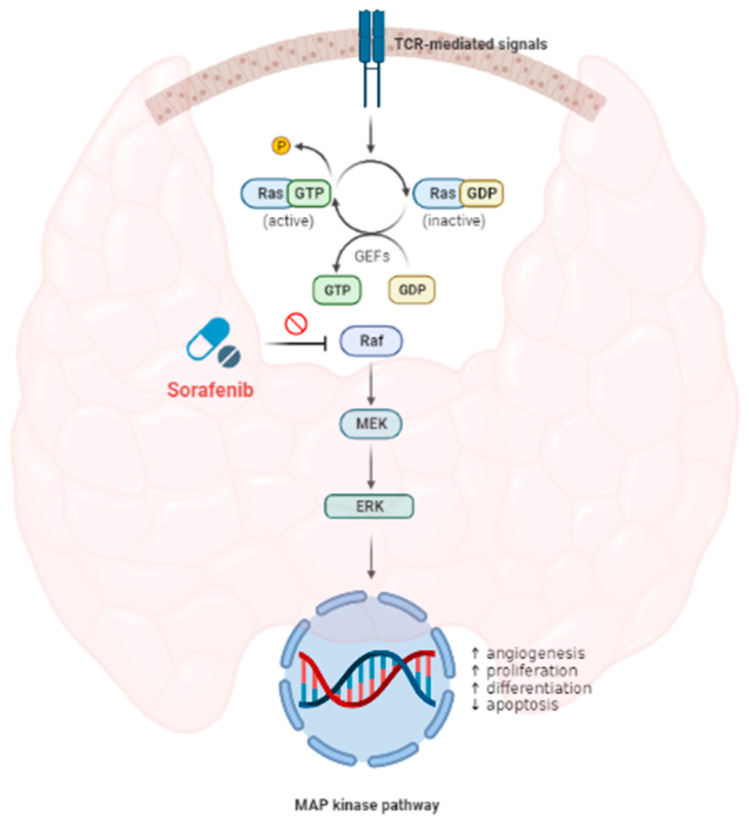
MAP kinase pathway and Sorafenib effect on cascade sequencing.

**Figure 2 medicina-58-00666-f002:**
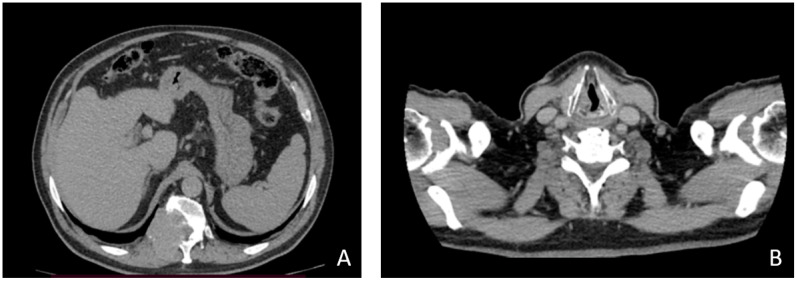
April 2021 CT Scan (**A**) Solid tissue infiltrating the D12 vertebral body; (**B**) neck scan with thyroidectomy outcomes.

**Figure 3 medicina-58-00666-f003:**
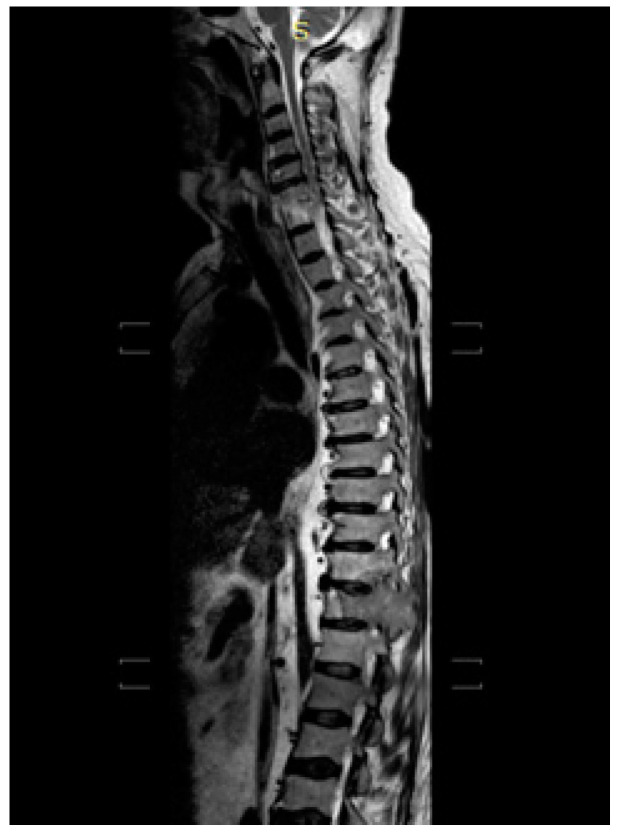
April 2021 MRI of the vertebral column: D12 is partially collapsed with morphostructural alteration and endocanal extrinsecation with considerable mass effect on medullary epiconus and signs of tissue suffering; tissue colonizes costal section of D12 and D11.

**Figure 4 medicina-58-00666-f004:**
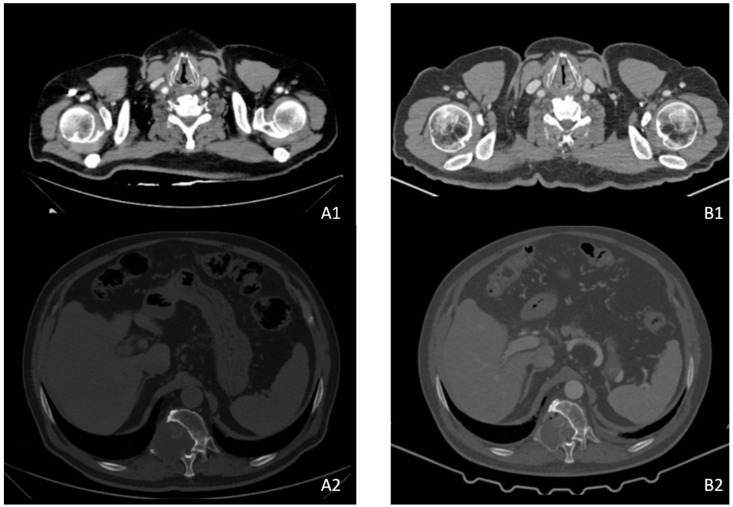
(**A1**,**A2**) Total-body CT scan August 2021; (**B1**,**B2**) total-body CT scan January 2022.

**Figure 5 medicina-58-00666-f005:**
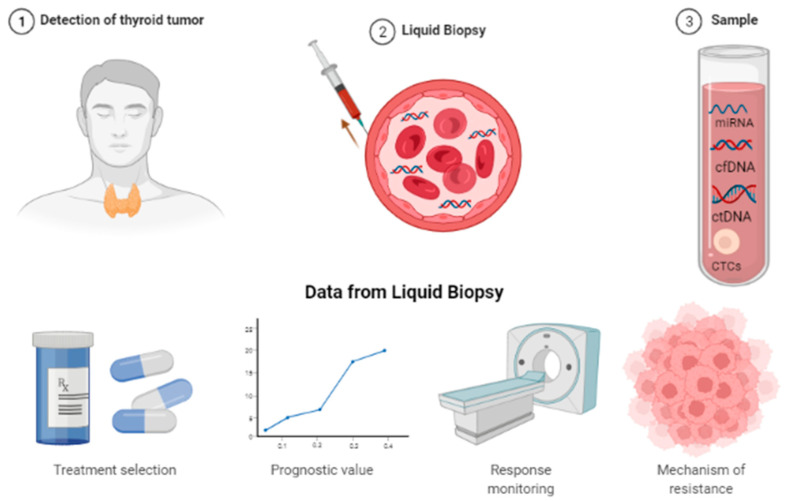
Benefit of liquid biopsy: treatment selection; prognostic value; response monitoring; mechanism of resistance.

## Data Availability

All data generated or analyzed during this study are included in this article. Further enquiries can be directed to the corresponding author.

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
