# Peer review of "Sorafenib in Metastatic Papillary Thyroid Carcinoma with BRAF K601E Mutation on Liquid Biopsy: A Case Report and Literature Review"

_medicina, 2022, doi:10.3390/medicina58050666_

Round 1

Reviewer 1 Report

The authors present a case of a 77 year old male with metastastic iodine refractory DTC (BRAF positive) who was treated with sorafenib leading to a good response. Further, the authors showed the usefulness of liquid biopsy to determine the mutational status of the tumor.

After reading the manuscript several important issues remain:

  1. General comment
  • Main comment on the current case report is that treating a patient with metastastic iodine refractory DTC with sorafenib is common practice, and there are multiple trials investigating this. It would have been more interesting if the patient was treated with targeted BRAF therapy like vemurafenib or dabrafenib. In my opinion the main addition of the current case report to literature is the fact that the mutational status of the tumor (BRAF positive) was obtained by liquid biopsy, without having a biopsy of the tumor mass. Therefore I would advise to emphasize this more in both the Abstract, Introduction, Discussion and Conclusion.
  1. Introduction
  • I would suggest to shorten the part about all kinds of mutations in thyroid cancer and introduce liquid biopsy.
  • Line 45-47. Please add appropriate references.
  • Line 85 ‘everyone’; please add appropriate references to back this very strong statement.
  • Please Introduce the goal of the case description, i.e. what do the authors want to illustrate with the case.
  1. Case Description
  • Line 107 ‘blood hypertension’; this could be changed into hypertension.
  • Line 110 ‘a total thyroidectomy surgery’; please remove surgery.
  • Line 115/116 ‘radiometabolic therapy’; is this RAI therapy?
  • Line 142; please change the typo
  1. Discussion
  • Line 181 to 184; please add appropriate references.
  • Line 270: why sorafenib was initiated and not lenvatinib?

Author Response

As suggested by the reviewer, we carried out minor revisions on English language, both in the highlighted lines and in other parts of the manuscript. We focused more on the use of liquid biopsy and shortened the introductory part about all mutations in thyroid cancer.
We added appropriate references as asked.
About the reason of initiating Sorafenib treatment instead of Lenvatinib, it is reported in line 297-300 "In the clinical case described by us, the presence of the BRAF mutation identified through the liquid biopsy and the possibility to choose only between two prescribed therapeutic options, Lenvatinib and sorafenib, made the therapeutic decision fall on sorafenib considering its inhibitory potential on BRAF." As there is no real and confirmatory comparison between the two drugs, keeping in consideration the differences between the two trials, we decided Sorafenib as written due to its inhibitory potential on BRAF

Reviewer 2 Report

Case report:

Sorafenib in Metastatic Papillary Thyroid Carcinoma with 2 BRAF K601E Mutation on Liquid Biopsy: A Case Report and 3 Literature Review

This article is informative because it describes in detail the genetic mutations in thyroid cancer and the genetic testing.

Comments to the Author

Although this paper is a case report, the introduction and discussion have the aspect of review article. The author should be further discussed for the treatment of this patient.

This case developed symptomatic bone metastasis a few months after surgery. Because the progression is too fast for papillary thyroid cancer, poorly differentiated carcinoma or anaplastic thyroid carcinoma is also suspected. Although genetic analysis indicated RAF mutation, was there no expression of TERT or TP53 mutation?

This patient underwent radiotherapy for bone metastases. Because of bone destruction and spinal cord compression, orthopedic treatment such as spinal fusion surgery should be considered. Surgical treatment for bone metastases should be considered. The authors need to consider in this point.

Sorafenib might have been selected because the genetic mutation in this patient had K601E BRAF, but lenvatinib is superior to sorafenib in tumor shrinkage effect and thus may be more effective in reducing patient symptoms. The authors need to consider how to select the TKIs.

This paper emphasizes the effectiveness of liquid biopsy. However, it is only about 4 months after surgery until the onset of bone metastasis. Genetic analysis from the pathology specimen may be more effective considering the accuracy of liquid biopsy. The authors need to discuss the reason for the choice of liquid biopsy.

Author Response

We followed up on the request of TERT or TP53 mutations (line 168-172).
Surgical treatment for bone metastases was considered after radiotherapy, as already stated in line 162. We added a few words to better clarify the therapeutic decision.
About the reason of initiating Sorafenib treatment instead of Lenvatinib, it is reported in line 297-300 "In the clinical case described by us, the presence of the BRAF mutation identified through the liquid biopsy and the possibility to choose only between two prescribed therapeutic options, Lenvatinib and sorafenib, made the therapeutic decision fall on sorafenib considering its inhibitory potential on BRAF." As there is no real and confirmatory comparison between the two drugs, keeping in consideration the differences between the two trials, we decided Sorafenib as written due to its inhibitory potential on BRAF
As asked by the reviewer, in line 245-252 we clarified why liquid biopsy was chosen to perform genetic analysis over the nonetheless recent pathology specimen.

Round 2

Reviewer 1 Report

I’m happy with the changes the authors made.

Author Response

There are no further concerns

Reviewer 2 Report

Comments to the author have been corrected.

If there are formalin-fixed specimens with a short fixation period, genetic analysis with fixed specimens would be preferred over liquid biopsy. Liquid biopsy is considered inferior to formalin-fixed specimens in detecting fusion genes such as NTRK fusion. Is the accuracy of analysis of genetic mutations by liquid biopsy comparable or inferior to that of fixed specimens? Please describe how much liquid biopsy costs and how long the test takes in Italy.

Author Response

We explained in further detail accuracy of liquid biopsy and the reason for our choice. Liquid biopsy for this patient cost 1500 euros and median turnaround time amount to around 10 days.